# Anti-Citrullinated Protein Antibody Reactivity towards Neutrophil-Derived Antigens: Clonal Diversity and Inter-Individual Variation

**DOI:** 10.3390/biom13040630

**Published:** 2023-03-31

**Authors:** Alexandra Cîrciumaru, Marcelo Gomes Afonso, Heidi Wähämaa, Akilan Krishnamurthy, Monika Hansson, Linda Mathsson-Alm, Márton Keszei, Ragnhild Stålesen, Lars Ottosson, Charlotte de Vries, Miriam A. Shelef, Vivianne Malmström, Lars Klareskog, Anca I. Catrina, Caroline Grönwall, Aase Hensvold, Bence Réthi

**Affiliations:** 1Division of Rheumatology, Department of Medicine Solna, Karolinska Institutet, Karolinska University Hospital, 17176 Stockholm, Sweden; 2Center for Rheumatology, Academic Specialist Center, Stockholm Health Services, 11365 Stockholm, Sweden; 3Thermo Fisher Scientific, 75450 Uppsala, Sweden; 4Department of Immunology, Genetics and Pathology, Uppsala University, 75185 Uppsala, Sweden; 5Department of Microbiology, Tumor and Cell Biology, Karolinska Institutet, 17165 Stockholm, Sweden; 6Division of Pediatric Endocrinology, Department of Women’s and Children’s Health, Karolinska Institutet, 17176 Stockholm, Sweden; 7Department of Medicine, University of Wisconsin-Madison, Madison, WI 53705, USA; 8William S. Middleton Memorial Veterans Hospital, Madison, WI 53705, USA

**Keywords:** rheumatoid arthritis, ACPA, neutrophil, NETosis, citrullination

## Abstract

Background: Why the adaptive immune system turns against citrullinated antigens in rheumatoid arthritis (RA) and whether anti-citrullinated protein antibodies (ACPAs) contribute to pathogenesis are questions that have triggered intense research, but still are not fully answered. Neutrophils may be crucial in this context, both as sources of citrullinated antigens and also as targets of ACPAs. To better understand how ACPAs and neutrophils contribute to RA, we studied the reactivity of a broad spectrum of RA patient-derived ACPA clones to activated or resting neutrophils, and we also compared neutrophil binding using polyclonal ACPAs from different patients. Methods: Neutrophils were activated by Ca^2+^ ionophore, PMA, nigericin, zymosan or IL-8, and ACPA binding was studied using flow cytometry and confocal microscopy. The roles of PAD2 and PAD4 were studied using PAD-deficient mice or the PAD4 inhibitor BMS-P5. Results: ACPAs broadly targeted NET-like structures, but did not bind to intact cells or influence NETosis. We observed high clonal diversity in ACPA binding to neutrophil-derived antigens. PAD2 was dispensable, but most ACPA clones required PAD4 for neutrophil binding. Using ACPA preparations from different patients, we observed high patient-to-patient variability in targeting neutrophil-derived antigens and similarly in another cellular effect of ACPAs, the stimulation of osteoclast differentiation. Conclusions: Neutrophils can be important sources of citrullinated antigens under conditions that lead to PAD4 activation, NETosis and the extrusion of intracellular material. A substantial clonal diversity in targeting neutrophils and a high variability among individuals in neutrophil binding and osteoclast stimulation suggest that ACPAs may influence RA-related symptoms with high patient-to-patient variability.

## 1. Introduction

The interaction of citrulline-reactive B cells and activated neutrophils is typically envisaged as a central mechanism in rheumatoid arthritis (RA) pathogenesis that, once triggered, may continue as a self-perpetuating driving force for chronic joint inflammation [1,2]. Citrullinated autoantigens released from cells that undergo NETosis could activate autoreactive B cells that would subsequently differentiate into plasma cells and produce anti-citrullinated protein antibodies (ACPAs), which directly, or via forming immune complexes, could induce further NETosis, thereby fueling further ACPA production and immune complex formation. Moreover, ACPAs, directly or via immune complexes formed with neutrophil-derived autoantigens, may also be crucial to activate other FcR-mediated mechanisms, stimulating inflammation, pain and bone loss [2,3,4,5,6,7]. The process of NETosis, i.e., the externalization of DNA–protein complexes from dying neutrophils, may also be important for understanding the well-known phenomenon that ACPAs frequently target proteins that are normally localized intracellularly, such as histones, ribonucleoproteins or cytoskeletal components [8,9]. 

NETosis is increased in RA, both at mucosal surfaces and in joints where the immunity against citrullinated and other post-translationally modified proteins is most likely initiated and later perpetuated [1,10]. Moreover, at least some forms of NETosis are mechanistically dependent on protein citrullination by the protein arginine deiminase (PAD)-4 enzyme, providing a further link towards the production of citrullinated autoantigens [11,12]. Although neutrophil activation has been shown to result in the generation of citrullinated autoantigens [1,13,14,15], the results are somewhat contradictory, as NETosis might occur without an increase in citrullination [16,17]. Moreover, a polyclonal anti-citrullinated vimentin antibody preparation has been shown to directly trigger NETosis [1], whereas certain monoclonal ACPAs have been shown to block NET production [18]. Notably, these inhibitory clones were also shown to reduce collagen antibody-induced arthritis in mice. A prerequisite of both the activation and inhibition of NETosis would be the binding of ACPAs to still-intact neutrophils, i.e., before citrullinated antigens could be released via NETosis itself, which suggests the presence of citrullinated antigens on the neutrophil surface, similar to what has been shown for osteoclasts [3,6]. 

The exact role for ACPA binding to neutrophil-derived antigens in RA is thus complex and not clearly understood. In the present work, we therefore decided to analyze in detail how ACPA binding to neutrophil-derived antigens is influenced by the pathways of cell activation, the location of target antigens (intra- or extracellular) and the activity of the PAD2 and PAD4 enzymes. From the antibody perspective, we studied the diversity among ACPA clones and the variability between affinity-purified polyclonal ACPAs from different patients in neutrophil binding. 

## 2. Materials and Methods

### 2.1. Neutrophil Isolation

Human neutrophils were isolated from the peripheral blood of healthy volunteers (*n* = 11), individuals at risk for RA from the Karolinska RiskRA cohort (*n* = 4) [19] and RA patients (*n* = 4) (Appendix A). The samples were collected with written informed consent and ethical approval by the Regional Ethics Review Board, Stockholm. Neutrophils were purified by using dextran sedimentation and a subsequent Ficoll separation (Ficoll-Paque PLUS; Cytiva Life Sciences, Uppsala, Sweden), followed by red blood cell lysis (0.2% NaCl solution, 40 s) as previously described [20]. 

Murine bone-marrow-derived neutrophils were isolated as described [21,22]. Wild type C57BL/6J mice were purchased from Charles River (Freiberg, Germany). PAD4^−/−^ mice (Cg-Padi4tm1.1Kmow/J, from Jackson Laboratory, Bar Harbor, ME, USA) and PAD2^−/−^ mice [23] backcrossed eight generations to the C57BL/6J background were bred at the Karolinska Institute. Experiments were performed with age- and sex-matched 11–16-week-old mice (*n* = 6 WT, *n* = 4 PAD4^−/−^ and *n* = 3 PAD2^−/−^ mice were used). All experimental procedures were approved by the regional ethical committee (Stockholms Djurförsöksetiska Nämnd). 

### 2.2. Neutrophil Activation 

Neutrophils were stimulated in RPMI medium (Thermo Fisher Scientific, Waltham, MA, USA) supplemented with 5% fetal calf serum (FCS) and glutamine (Sigma-Aldrich, St. Louis, MO, USA) using 25 μM A23187 (if not otherwise specified in the text), 0.2 μM phorbol 12-myristate 13-acetate (PMA), 0.5 μM nigericin (all from Sigma-Aldrich), 10 μg/mL zymosan (InvivoGen, San Francisco, CA, USA), 25 ng/mL IL-8 (Peprotech, London, UK) or 50 μg/ml ACPAs. The PAD4 inhibitor BMS-P5 [24] was applied at 10 μM concentration, room temperature, 30 min prior to activation. The IncuCyte ZOOM platform (Essen BioScience, Hertfordshire, UK) was used for NET imaging as previously described [20]. Briefly, the total cell number was established by counting NUCLEAR ID Red (Enzo Life Sciences, Farmingdale, NY, USA)-positive areas < 15 μm^2^, whereas SYTOX green-positive areas > 100 μm^2^ were considered as NETs and were counted at consecutive time points.

### 2.3. Osteoclast Cultures

Osteoclast cultures were established from human peripheral blood monocytes as described previously [25]. 

### 2.4. ACPA and Control Antibodies

Monoclonal ACPAs were cloned from synovial plasma cells (1325:01B09, 1325:04C03, 1325:05C06) [9], blood memory B cells (37CEPT2C04, 37CEPT1G09, 37CEPF1C40, BVCA1) [26,27], bronchoalveolar lavage B cells (L204:01A01, L201:11C11, L204:05E10, L201:10D07, Joshua V et al. Manuscript under review) or bone marrow plasma cells (254:17D08 and 254:C7 × 1604, Hensvold A. et al. Manuscript under review). All antibodies were expressed as recombinant human IgG1 antibodies, as described previously [9,26,27,28], similarly to the control antibody clone 1362:01E02 (with no known reactivity to any post-translationally modified or native antigens), which was derived from a synovial memory B cells from an ACPA-negative RA patient. Polyclonal ACPA and an ACPA-depleted control IgG were purified from pooled or non-pooled sera of RA patients, as described previously [6,29,30]. All antibody preparations were extensively quality controlled, with endotoxin levels and protein aggregation assessed as previously described [28]. The concentrations of the antibody preparations were determined by ELISA [28]. For antibody biotinylation we used the Lightning-Link Rapid Biotin Antibody Labeling Kit (Novus Biologicals, Centennial, CO, USA).

### 2.5. Fine-Specificity Analysis of the Tested Antibodies

The bindings of monoclonal ACPA to citrullinated, acetylated and carbamylated peptides were investigated at 5 μg/mL IgG using commercial CCP2 ELISA (CCPlus immunoscan, Svar Life Science, Malmö, Sweden), as well as modified-vimentin peptide ELISA (Mod-Vim_58–69_, Organtec Diagnostics, Mainz, Germany) and a custom-designed antigen microarray (Thermo Fisher Scientific, ImmunoDiagnostics, Uppsala, Sweden), as previously described [8]; see Appendix A for peptide details. In addition, bindings to citrullinated histone 4 (Cit-His4_14–34_, Cit-His4_1–18_), citrullinated heterogeneous nuclear ribonucleoprotein A1 (Cit-hnRNPA_199–212_, Cit-hnRNPA_211–224_) and acetylated histone 2B (Acet-His2B_6–22_) were studied using in-house ELISA as previously described [26]. Briefly, biotinylated peptides were captured on streptavidin-coated high-capacity plates (Thermo Fisher Scientific) and antibody binding was assessed at 5 μg/mL in RIA buffer (1% BSA, 325 mM NaCl, 10 mM Tris-HCl, 1% Tween-20, 0.1% SDS) and detected with HRP Fab’2 goat anti-human IgG (γ) (Jackson Immunoresearch, Ely, UK). Bindings in all assays were compared to native lysine- or arginine-containing peptides. Plasma samples or polyclonal ACPA IgG preparations (5 μg/mL) were analyzed using antigen microarray as described above.

### 2.6. Flow Cytometry

The cells were either directly labelled with the above-noted antibodies using PBS with 2.5% FCS for staining and washes or were fixed, permeabilized and then stained using the Fixation/Permeabilization kit (BD Biosciences, Stockholm, Sweden) following the manufacturer’s protocol. To reduce non-specific antibody binding, the FcR Blocking Reagent (human or murine) of Miltenyi Biotech was used prior to staining. Antibodies were added to the samples (2 μg in 50 μL) for 30 min at 4 °C, then the samples were washed and incubated with APC-labeled anti-human IgG (BD Biosciences) for 30 min at 4 °C. After two subsequent washes, the samples were fixed in 1% paraformaldehyde and stored at 4 °C prior to analysis. To evaluate neutrophil death, we used Live/Dead fixable dead cell staining (Thermo Fisher), and in some experiments SYTOX Green (Thermo Fisher) or BV421-labeled Annexin-V (BD Biosciences). We studied degranulation using AlexaFluor 488-labelled anti-CD107a (Biolegend, San Diego, CA, USA). The cells were analyzed on a FACSVerse flow cytometer, with a gating strategy depicted on Appendix A, and the data were analyzed using FlowJo software v10.8.1(BD Biosciences). 

### 2.7. Immunofluorescence

Neutrophils were fixed in 2% paraformaldehyde (15 min), washed twice in PBS and the slides were then dried and stored at −20 °C. For staining, the cells were permeabilized using 0.1% TritonX-100 (Sigma-Aldrich) in PBS twice for ten minutes at room temperature, then blocked with 2% human AB serum (Clinical Immunology, Academic University Hospital, Uppsala, Sweden) in PBS for 30 min at room temperature, washed three times in PBS and then incubated with biotinylated monoclonal ACPAs or control antibodies (10 μg/mL) in a humid chamber overnight at 4 °C. After three washes in PBS, streptavidin-AlexaFluor488 (BioLegend) was added for 1 h at room temperature. DNA was stained with DAPI (Sigma-Aldrich) and images were acquired with a Zeiss LSM 880 confocal microscope and further analyzed using ImageJ 1.53t software (Wayne Rasband and contributors, National Institute of Health, Bethesda, MD, USA).

### 2.8. Statistical Analysis

Differences between sample groups were analyzed by using two-way analysis of the variance, followed by Dunnett’s multiple comparison tests using GraphPad Prism 9 (GraphPad Software Inc. San Diego, CA, USA). *p* values < 0.05 were considered significant.

## 3. Results

### 3.1. ACPA Binding to NET-Associated Antigens and Nuclear Epitopes in Intact Cells

To understand how different activation pathways acting via unique receptors and signaling mechanisms influence ACPA binding to neutrophils, we exposed human peripheral blood neutrophils to a set of different stimuli, including PMA, nigericin, the calcium ionophore A23187, IL-8, the TLR2 ligand Zymosan or pooled polyclonal human ACPA and control IgG preparations. PMA, nigericin and A23187 induced NETosis (detected as expanded, >100 μm^2^ SYTOX green-positive area in IncuCyte), whereas cells treated with other stimuli or left untreated preserved their viability for several hours (Figure 1A). A flow cytometric analysis of cell size and morphology revealed a severe contraction and granularity loss in the presence of A23187 (Figure 1B), and to a much lesser extent with PMA, indicating a partial discrepancy between microscopic- and flow cytometry-based analyses, probably due to the fact that NETs without particulate, cell-like structures were not detected by flow cytometry (e.g., in the presence of nigericin). 

The microscopic analysis of the cells revealed a typical segmented, multi-lobulated nuclear structure of living neutrophils, with only a few sporadic NETs in the untreated and in the IL-8- or Zymosan-treated cultures (Figure 1C). In contrast, the Ca^2+^ ionophore-treated neutrophils were characterized by a disrupted nuclear DNA structure and somewhat spread DNA-rich area as compared with intact neutrophils, but a relatively well-preserved particulate shape, which may explain the good recovery of disrupted cellular structures in flow cytometry. PMA and nigericin induced a more filamentous, expanded, carpet-like NET release with few cell-shaped structures remaining, in line with the higher detected NETosis rate as compared with A23187 (Figure 1A). 

Next, neutrophils exposed to the different stimuli were stained with two well-characterized monoclonal ACPAs, 1325:04C03 (C03) and 1325:01B09 (B09) [8,9], or with the control IgG clone 1362:01E02, and analyzed with confocal microscopy (Figure 1C). Notably, B09 labelled the chromatin within the intact nuclei and also several of the NET structures, whereas C03 bound to several of the NETs but not to intact nuclei. PMA, which may induce relatively low levels of citrullination [16,17], resulted in less ACPA binding to the extruded material as compared with the other stimuli. 

When evaluated by flow cytometry, only cells treated with a high dose of Ca^2+^ ionophore were targeted, with strong binding by C03 or a polyclonal ACPA preparation and weak/partial binding by B09 (Figure 1D). Notably, lower Ca^2+^ ionophore concentrations, associated with less cell damage (Figure 1B), resulted in lower ACPA binding to the cells (Figure 1D). The lack of ACPA binding to PMA- and nigericin-stimulated cultures is likely to reflect a limited recovery of the outspread filamentous NET structures in these samples by flow cytometry. Moreover, the relatively weak and partial chromatin staining by B09, seen in the intact cells after permeabilization (Figure 1C), did not reach the sensitivity threshold of flow cytometry (Figure 1D).

### 3.2. ACPA Targets Are Produced in Activated Neutrophils and Show Little Accessibility to Antibodies

As we have detected a preferential binding of ACPAs to NET structures and to nuclear antigens, which are non-accessible in intact cells (Figure 1C), we analyzed in detail whether intact neutrophils could be targeted by ACPAs. It is noteworthy that live and dead cell identification is not trivial in neutrophils, as the staining of dead cells can be compromised by NETosis itself due to the release of DNA and other intracellular content. Neutrophils with intact cellular and nuclear morphologies were undetectable in A23187, PMA or nigericin-treated cultures when observed by using microscopy (Appendix A). In contrast, the percentage of dead cells determined by Live/Dead, SYTOX Green or Annexin-V staining remained unexpectedly low in the flow cytometry data. Similarly, degranulation frequencies measured via the staining of CD107 remained very low in the presence of PMA and ionomycin. Due to the fact that we found the data obtained by using different types of dead cell markers inconsistent with the microscopic view of the cells, we relied more on the visual signs of cell damage, disintegrated nuclei and the spreading of DNA when distinguishing samples with intact or disrupted cells.

To investigate the accessibility of ACPA targets, we stained non-treated and Ca^2+^ ionophore-activated neutrophils with a broad range of monoclonal ACPAs, both with and without fixation and permeabilization. We could detect a variable level of ACPA binding to the cells, but overwhelmingly in samples that were both activated and fixed/permeabilized, suggesting that ACPA binding requires cellular activation and that the citrullinated autoantigens remain non-accessible for antibodies in the extracellular space, at least during the time frame of the experiment (Figure 2A,B). 

The tested ACPA clones originated from different anatomical locations (Figure 2B), and although the spectra of autoantigens might be different at these sites, the source of antibodies did not have a major impact on the binding to neutrophil-derived antigens. Notably, the tested ACPA clones were characterized by distinct antigen-recognition patterns (Figure 2C), which may explain their differential binding to neutrophils. Consistent with the flow cytometry data, ACPA binding could be detected in confocal microscopy in pre-fixed and permeabilized cells, but not when the staining was performed before fixation (Figure 2D). Our results thus suggest that activation signals that lead to immense cell death are needed for the production of target antigens in neutrophils and that these antigens have limited accessibility during the first hours of NETosis. In line with these findings, the co-incubation of neutrophils with various ACPA clones failed to induce NETosis and, moreover, it did not influence the kinetics or extent of NETosis triggered by A23187 (Figure 2E). To understand whether neutrophils might be more primed to bind ACPAs in conditions associated with chronic immune activation, we stained intact neutrophils freshly isolated from individuals at risk of RA or from RA patients with various ACPA clones. Sporadically, we observed a slightly increased binding of some of the ACPAs as compared with the control antibodies, although the labeled cells were typically dead, suggesting further that ACPAs primarily target damaged neutrophils (Appendix A).

### 3.3. PAD4-Dependent Recognition of Activated Neutrophils by ACPAs

We analyzed whether different ACPA clones were characterized by a bias towards PAD2- or PAD4-dependent targets using bone-marrow-derived neutrophils from wildtype (C57BL6) or PAD2- or PAD4-deficient mice. Notably, whereas the lack of PAD2 impacted minimally, or not at all, the binding of ACPAs to neutrophil-derived antigens, most of the tested clones and the polyclonal ACPA preparation bound to neutrophils in a largely PAD4-dependent manner (Figure 3A,B). Two of the tested clones, C04 and G09, which originated from the same phylogenetically related B cell clade [27], were characterized by a substantial PAD4-independent neutrophil binding capacity. Consistent with this finding, when ACPA binding was studied in human neutrophils pretreated with the highly selective PAD4 inhibitor BMS-P5, a strongly reduced binding of several of the clones and of the polyclonal ACPA preparation was observed, with the exception of clones C04, C06 and G09 that preserved a relatively high PAD4-independent binding capacity (Figure 4). 

### 3.4. ACPA-Positive Individuals Differ in ACPA Fine Specificities and Cell-Targeting Antibodies

The clonal diversity in ACPA binding to neutrophil-derived antigens (Figure 1 and Figure 2) may suggest a high patient-to-patient variability in the capacity of ACPAs to target neutrophils. To test this, we isolated ACPAs from the serum of eight different ACPA-positive RA patients for functional analyses. In spite of a presumably broad ACPA repertoire in the circulation, the fine-specificity analysis revealed striking differences between individuals (Figure 5A), with some specificities presented in single patients only (e.g., citrullinated tenascin-1 binding in patient 116), whereas citrullinated fibrinogen or filaggrin were targeted in many of the individuals. One of the patients lacked all tested ACPA fine-specificities (though carrying antibodies that reacted with CCP2). When we analyzed the binding to neutrophil-derived antigens by the individual ACPA preparations, we observed a clearly heterogenic binding capacity, with more or less neutrophil targeting in different individuals (Figure 5B). In fact, ACPAs from two patients did not detectably bind to neutrophils at all. None of the antibody preparations induced NETosis (Appendix A). Importantly, if neutrophil targeting by ACPAs carries pathological relevance in RA, our results indicate that this mechanism may be present in some of the ACPA-positive individuals, but not, or weakly, in others. To test further whether individual differences in the ACPA repertoire could influence ACPA functions, we turned to an unrelated assay in which the cellular action of ACPAs has been previously demonstrated [3,6,9,31], and we tested osteoclast differentiation in the presence of the ACPA preparations obtained from the eight different patients (Figure 5C). Notably, only two of the eight individuals carried ACPAs with a capacity to stimulate osteoclasts in cell culture, with only one reaching statistical significance, further suggesting that ACPAs might influence disease pathways with high patient-to-patient variability. 

## 4. Discussion

In the present study we aimed to understand better how ACPAs target neutrophils, a potentially important cell type in RA pathogenesis [32]. In response to certain stimuli, neutrophils readily expel their DNA in complex with intracellular proteins, which subsequently form NETs that can immobilize microbes in the extracellular space. Citrullinated proteins have been detected in the NETs, most likely due to PAD4 activation during NETosis and/or the release of soluble or NET-associated PAD4 by the activated neutrophils [1,13,14,15]. Neutrophils themselves have been shown to react to ACPAs, with increased NETosis when exposed to polyclonal anti-citrullinated vimentin antibodies [1], whereas certain monoclonal ACPAs inhibited NETosis [18]. Such diverse effects of different ACPA preparations may suggest a high clonal variability in targeting neutrophils.

One of our aims was to detect and distinguish ACPA-binding to intact neutrophils versus their extruded material. We found that ACPAs broadly targeted NET-like structures, irrespective of whether these appeared only sporadically (as in untreated or IL-8 or Zymosan-treated cultures) or in high numbers (in A23187, PMA or nigericin-activated cultures). In contrast with damaged cells, ACPAs did not bind to intact cells and did not stimulate or inhibit NETosis. By using confocal microscopy, we could detect faint chromatin binding by the ACPA clone B09, but not with the ACPA clone C03, in cells with intact nuclear morphology. These observations suggest that at least some ACPA targets may be present within intact cells, although not exposed to the antibodies present in the extracellular space. Even in Ca^2+^ ionophore-activated cells, the detection of ACPA binding required the fixation and permeabilization of the samples, suggesting that the citrullinated antigens remained hidden from the antibodies, at least during early phases of NETosis. Plausibly, these antigens would be exposed later, upon further disintegration of the extruded material. Overall, these results thus indicate that ACPAs recognize citrullinated antigens extruded from dying neutrophils under various conditions, whereas no binding to intact cells was detected.

In addition to citrullinated epitopes, ACPA clones (including several of the tested antibodies) can typically react with peptides subjected to other types of post-translational modifications as well [8,9]. Here, we showed that most of the ACPA clones and a pooled polyclonal ACPA preparation required PAD4, but not PAD2, for binding to neutrophil-derived antigens. Although PAD4 deficiency might influence NETosis itself and reduce the exposure of citrullinated antigens indirectly, it is important to note that we detected ACPA binding only in permeabilized samples, even in the presence of functional PAD4 enzymes. The role of PAD4 may therefore be crucial in the production, rather than in the exposure, of citrullinated antigens. The only two ACPA clones with substantial PAD4-independent binding capacities (C04 and G09) were phylogenetically related [27], further suggesting that a PAD4-independent binding capacity to neutrophil-derived antigens exists, but may be a relatively rare feature among the ACPA clones. This was further corroborated from the observation of the predominantly PAD4-dependent binding of polyclonal ACPAs. 

To envisage how ACPAs modulate pathways in RA pathogenesis, it could be instrumental to analyze and better understand the impact of clonal variability, as different types of ACPA effects can be expected in patients carrying different ACPA repertoires. Importantly, by using a large repertoire of monoclonal ACPAs here, we presented a substantial clonal variability in the recognition of neutrophil-derived antigens, indicating that neutrophils are targeted by several of the ACPA clones but, at the same time, are not uniform targets for all ACPAs. 

Clonal differences have also been described among ACPAs in inducing osteoclast differentiation [9], fibroblast migration [30], pain and joint inflammation [27] or in blocking collagen-antibody-induced arthritis in mice [18,33]. Such clonal diversity in the antibody effects suggested that ACPA-positive individuals with diverse autoantibody repertoires may carry autoantibodies with very different capacities to bind certain cellular targets. Indeed, we observed a large variability in binding to neutrophil-derived targets among polyclonal ACPAs isolated from different RA patients. Notably, the concept of patient-to-patient variability in targeting neutrophil-antigens have also emerged in a previous study measuring serum immunoglobulin reactivity to NET components [34], although in this study ACPAs were not analyzed and predominantly non-citrullinated autoantigens were detected. 

Similar to the neutrophil data, here we also showed that polyclonal ACPA preparations obtained from eight different RA patients had very different capacities to stimulate osteoclast differentiation in vitro. Only two out of the eight individuals carried ACPAs with the overall capacity to stimulate these cells, and notably, we did not detect any association between targeting neutrophils and osteoclasts by the antibody preparations. In fact, by studying only two different cellular targets and eight ACPA-positive RA patients, we could already identify four different scenarios: ACPAs targeting either osteoclasts or neutrophils, ACPAs targeting both cell types or neither of these. We found these results particularly interesting, as they may imply a rather stochastic role for ACPAs in RA pathogenesis, largely determined by the compositions of the ACPA repertoires in different individuals. 

In conclusion, our study confirmed the link between NETosis and ACPA binding to neutrophil-derived antigens under various experimental conditions. Importantly, the capacity to bind neutrophils differed immensely between ACPA clones and, as a likely consequence of this, ACPAs isolated from patients with diverse clonal repertoires also bound neutrophil-derived antigens with high variability. Specificities that allow for the targeting of different cell types may appear stochastically during the course of the disease and, consequently, ACPAs may trigger unique combinations of pathways and patterns of symptoms in different patients. 

## Figures and Tables

**Figure 1 biomolecules-13-00630-f001:**
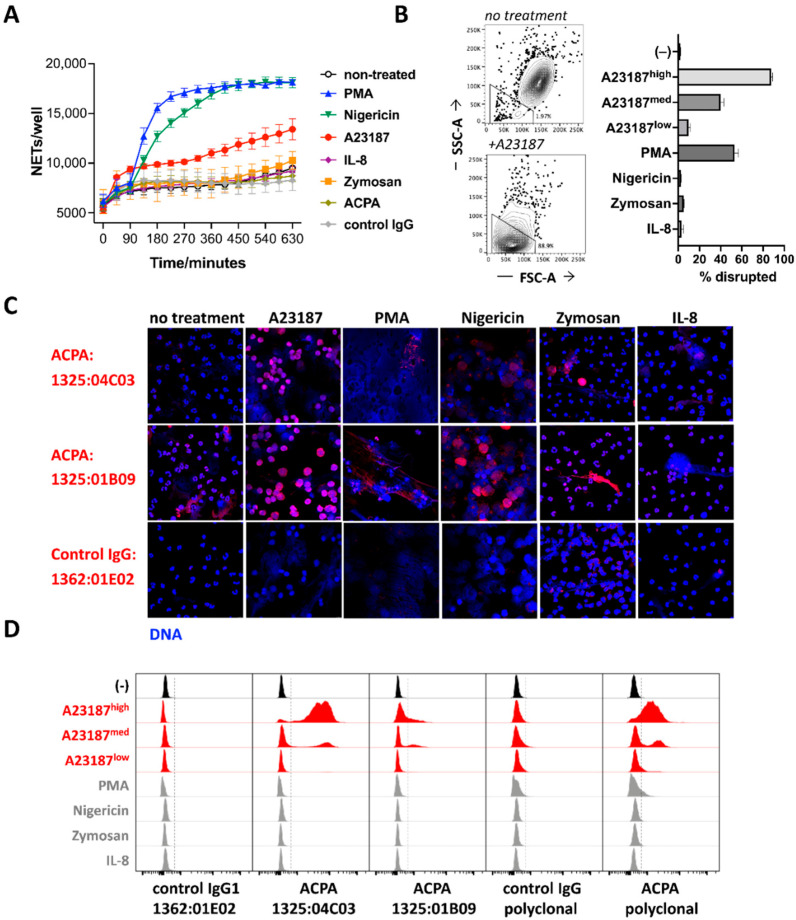
ACPA binding to neutrophil-derived antigen is associated with NETosis. Neutrophils were exposed to different stimuli and NETosis was quantified using IncuCyte (**A**). Granularity and cell size were analyzed by flow cytometry (**B**). Neutrophils were activated by using different stimuli for 3 h, and binding of ACPA or control antibodies was studied by confocal microscopy. DNA was stained by DAPI (**C**). ACPA binding was also analyzed by flow cytometry under the same conditions, using monoclonal and polyclonal ACPA and control antibody staining on fixed and permeabilized samples. Dashed lines, placed with the help of the control antibody-stained samples, represent the cutoff for stained events (**D**). Representative results of three independent experiments are shown. A23187^high^, A23187^med^ and A23187^low^ concentrations correspond to 25, 5 and 1 μM, respectively.

**Figure 2 biomolecules-13-00630-f002:**
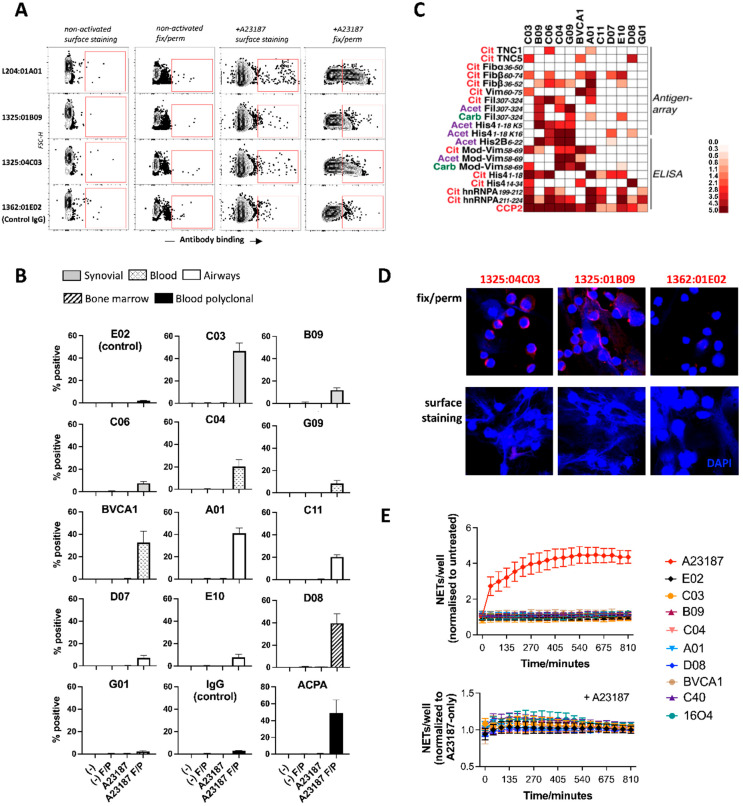
ACPA binding to intracellular epitopes in activated neutrophils. ACPA binding was studied to intact or fixed/permeabilized neutrophils that were either non-activated or exposed to A23187 for 45 min. Representative contour plots are shown with gated positive events (**A**) or cumulative data from three independent experiments, with staining detectable in activated and fixed/permeabilized (F/P) neutrophils only (**B**). Fine-specificity of the tested clones was analyzed to citrullinated (Cit), carbamylated (Carb) and acetylated (Acet) autoantigens (**C**). ACPA binding was analyzed by confocal microscopy to A23187-activated neutrophils in fixed/permeabilized or non-fixed samples (**D**). The effect of ACPA and control antibodies on NETosis was analyzed using IncuCyte; results are expressed after normalization to non-treated samples ((**E**), upper panel). Alternatively, the cells were exposed to a moderate dose (5 μM) of A23187 with or without the different antibodies, and values were normalized to samples treated with A23187 only ((**E**), lower panel). Average values calculated from three independent experiments are show.

**Figure 3 biomolecules-13-00630-f003:**
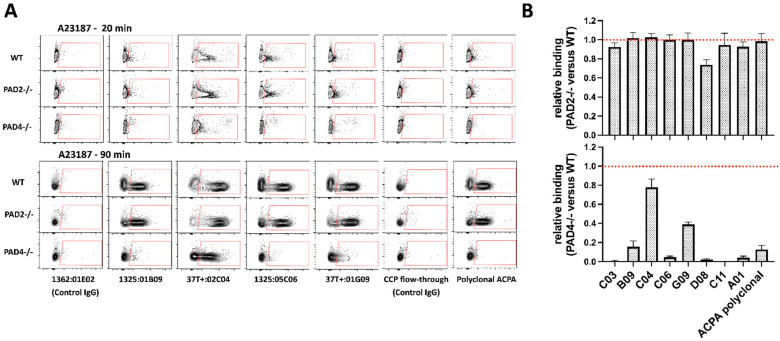
ACPA binding to PAD2- or PAD4-deficient neutrophils. Neutrophils were isolated from wild-type, PAD2- or PAD4-deficient mice, activated with A23187 and then fixed/permeabilized and stained with different ACPA and control antibody preparations. Representative stainings are shown with contour plots, with antibodies characterized by more or less PAD4-independent binding capacity (**A**). Average values are shown from at least three independent experiments (**B**). PAD4 deficiency resulted in a significant decrease in neutrophil binding (*p* < 0.001 for all clones, except for C04 where *p* < 0.01).

**Figure 4 biomolecules-13-00630-f004:**
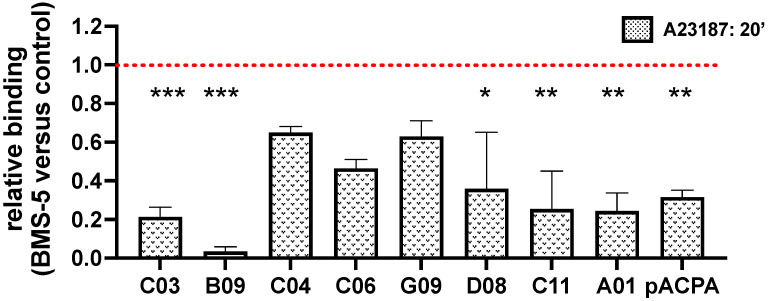
The PAD4 inhibitor BMS-5 reduces ACPA binding to neutrophil-derived antigens Human peripheral blood neutrophils were pre-treated with the BMS-5 or DMSO for 30 min at room temperature, followed by their activation with A23187. The cells were thereafter fixed, permeabilized and stained with different monoclonal or polyclonal ACPA preparations. Average values are shown, calculated from at least three independent experiments. *** *p* < 0.001, ** *p* < 0.01, * *p* < 0.05.

**Figure 5 biomolecules-13-00630-f005:**
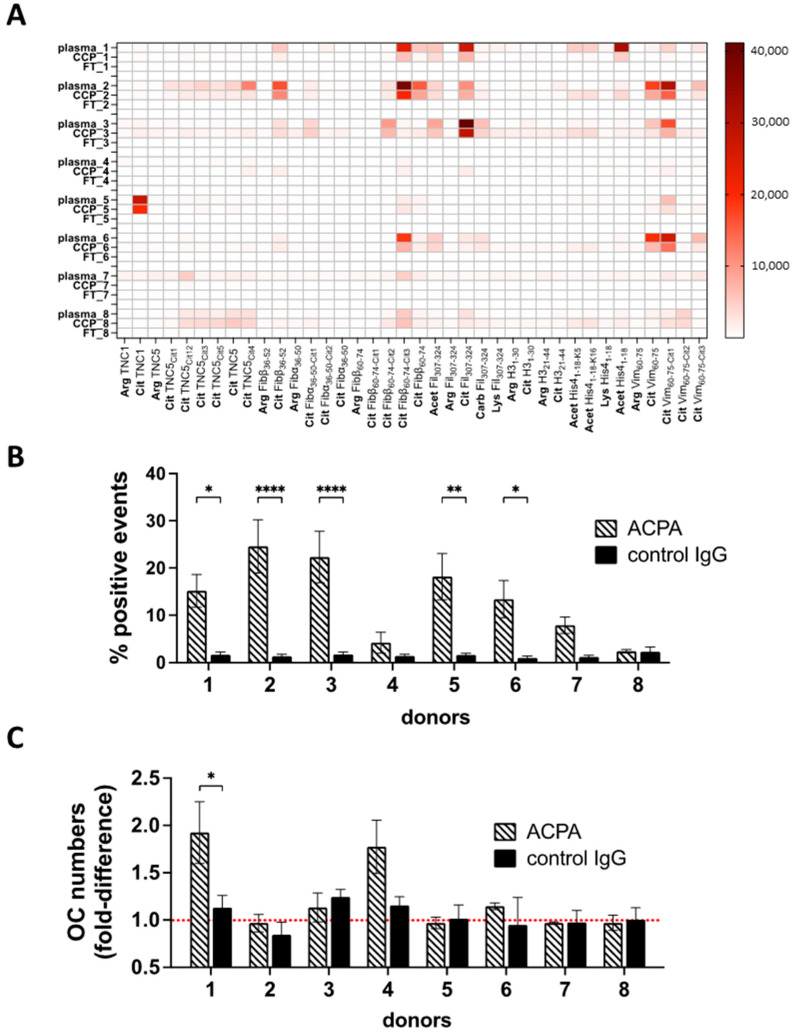
Individual variability in ACPA cellular targets. ACPAs were purified from plasma samples obtained from eight individual RA patients. Fine-specificity patterns of the original plasma samples, ACPAs purified with CCP affinity chromatography and the ACPA-depleted IgG fractions (flowthrough, FT) were studied using a multiplex solid-phase peptide array platform (**A**). Peripheral blood neutrophils were activated with A23187 for 45 min, then fixed and permeabilized and stained with the different ACPA or the corresponding control (ACPA-depleted) IgG preparations (**B**). The same antibodies were also added to cultures of developing OCs. Osteoclast numbers were normalized to control samples cultured without antibodies; average values were calculated from three independent experiments (**C**). **** *p* < 0.0001, ** *p* < 0.01, * *p* < 0.05.

## Data Availability

Data are available from corresponding author upon request.

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
