# Peer review of "Anti-Citrullinated Protein Antibody Reactivity towards Neutrophil-Derived Antigens: Clonal Diversity and Inter-Individual Variation"

_biomolecules, 2023, doi:10.3390/biom13040630_

Round 1

Reviewer 1 Report

In this paper, the authors studied the binding of a number of ACPA antibody clones derived from RA patients to activated or resting neutrophils.  Overall, they show that there is a high variability among the binding capacity of the antibodies. This is an interesting information to help understand the role of ACPA antibodies in RA.  I think it would be useful to see more clearly how the cell disruption (as measured by size and granularity) is associated with production of NETosis, in addition to the IncuCyte data. Did the authors try to examine the induction of NETosis with SYTOX stain or similar dyes using flow cytometry? 

Other points:

Material and methods - Flow cytometry – ‘Antibodies were added to the samples (2ug in 50ul)’ – please clarify how the amount of Ab was determined

On Fig 1A, the “A23187” is high, medium or low concentration? Same for Fig 1C. Also, please provide a summary of the IncuCyte read out.

Fig 1D shows binding of ACPA to activated neutrophils – was it done after fixation and permeabilization? Please show the gating that generate the data for the histograms .

Again, for Fig 2A, please show the gating.

Fig 2B: in my PDF, for the legend, both synovial and airways rectangles are white, while the figure has grey and white. Please correct the legend.

Fig 2C – the antigen array and the ELISA are not described in the methods section. Similarly for Fig 5A .

What’s the purity of the isolated neutrophils?

Author Response

In this paper, the authors studied the binding of a number of ACPA antibody clones derived from RA patients to activated or resting neutrophils.  Overall, they show that there is a high variability among the binding capacity of the antibodies. This is an interesting information to help understand the role of ACPA antibodies in RA.  I think it would be useful to see more clearly how the cell disruption (as measured by size and granularity) is associated with production of NETosis, in addition to the IncuCyte data. Did the authors try to examine the induction of NETosis with SYTOX stain or similar dyes using flow cytometry? 

Thank you for the comments.

The comparison of different dead cell markers with the microscopic view of the cells (and DNA) and the flow cytometry-based visualization of cell disruption is shown in Supplementary figure 1 and discussed in the second ‘Results’ chapter. Please note that SYTOX green, in itself, is a DNA dye that stains all dead cells that contain DNA. A size threshold (>100mm2 ) was introduced in the IncuCyte assay to separate SYTOX green-positive NETs from the (smaller) SYTOX green-positive cells that died through other mechanisms (based on reference 20). We found the collapse of FSC/SSC parameters in Ca2+ ionophore activated neutrophils interesting because it showed that the binding of ACPAs correlated with cell damage, i.e. different concentrations of the Ca2+ ionophore A23187 primed neutrophils for ACPA binding proportionally to the level of cell damage. We added a sentence to the first Results chapter:

Notably, lower Ca2+ ionophore concentrations, associated with less cell damage (Figure 1B), resulted in lower ACPA binding to the cells (Figure 1D).

Other points:

Material and methods - Flow cytometry – ‘Antibodies were added to the samples (2ug in 50ul)’ – please clarify how the amount of Ab was determined

Antibody levels were determined by ELISA, which is described in detail in reference 28. We added a sentence to the methods section:

Concentration of the antibody preparations was determined by ELISA [28].

On Fig 1A, the “A23187” is high, medium or low concentration? Same for Fig 1C. Also, please provide a summary of the IncuCyte read out.

Thank you for pointing out this missing information. In the figure legend the following sentence is added:

A23187high, A23187med and A23187low concentrations correspond to 25, 5 and 1uM, respectively.

In the methods, it is now stated:

Neutrophils were stimulated in RPMI medium (Thermo Fisher Scientific, Waltham, Massachusetts, USA), supplemented with 5% fetal calf serum (FCS) and glutamine (Sigma-Aldrich, St. Louis, MO, USA), using 25uM A23187 (if not otherwise specified in the text)...

About the IncuCyte method, we added the following sentence to the methods section:

Briefly, total cell number was established by counting NUCLEAR ID Red (Enzo Life Sciences, Farmingdale, NY, USA) positive areas <15 mm2, whereas SYTOX green positive areas >100mm2 were considered as NETs, and were counted at consecutive time points.

Fig 1D shows binding of ACPA to activated neutrophils – was it done after fixation and permeabilization? Please show the gating that generate the data for the histograms . Again, for Fig 2A, please show the gating.

We complemented the legend of Figure 1 as follows:

ACPA binding was also analyzed by flow cytometry under the same conditions, using monoclonal and polyclonal ACPA and control antibody staining on fixed and permeabilized samples.

For the basic gating strategy, we prepared a general figure (Supplementary figure 2.) for human and murine cells. Examples for gating for ACPA-positive cells are shown on Figure 1D, Figure 2A, Figure 3A.

We complemented the Flow cytometry section in Methods, as follows:

The cells were analyzed on a FACSVerse flow cytometer, with a gating strategy depicted on Supplementary figure 2, and the data were analyzed using FlowJo software (BD Biosciences).

Fig 2B: in my PDF, for the legend, both synovial and airways rectangles are white, while the figure has grey and white. Please correct the legend.

The synovial antibodies are marked with grey colour. I higher resolution image is uploaded separately.

Fig 2C – the antigen array and the ELISA are not described in the methods section. Similarly for Fig 5A .

Thank you for noticing this. We added a paragraph to the methods section and provide a supplementary table with the peptide sequences.

What’s the purity of the isolated neutrophils?

Based on the FSC/SSC characteristics and the purity was 97-99% for human cells and 88-91% in murine preparations, which was also in line with the microscopic view of the cells (Figure 1). In other projects, we observed very homogenous expression patterns in FcgRII and FcgRIII or CXCR1 and CXCR2 among human neutrophils gated based on high SSC values, suggesting a homogenous population (see the uploaded 'coverletter' image). The lower purity in murine cell preparation was partially compensated by a more restrictive baseline gating, as depicted on Supplementary figure 2.

Expression of FcgRs and chemokine receptors on neutrophils gated via FSC/SSC parameters. The grey histograms represent isotype control stainings.

Expression of FcgRs and chemokine receptors on neutrophils gated via FSC/SSC parameters. The grey histograms represent isotype control stainings.

Reviewer 2 Report

1. This article deals with how neutrophils and ACPAs work in the pathogenesis of RA and suggests clonal diversity in targeting neutrophils and variability by individuals in neutrophil binding may influence the variability of RA-related symptoms. The findings are meaningful and well-written, except for some minor points to be revised.

2. Please indicate the number of healthy volunteers and patients with RA who participated as well as the number of experimental murine used in this study. Additionally, it would be helpful to show the baseline characteristics of the participants.

Author Response

  1. This article deals with how neutrophils and ACPAs work in the pathogenesis of RA and suggests clonal diversity in targeting neutrophils and variability by individuals in neutrophil binding may influence the variability of RA-related symptoms. The findings are meaningful and well-written, except for some minor points to be revised.
  2. Please indicate the number of healthy volunteers and patients with RA who participated as well as the number of experimental murine used in this study. Additionally, it would be helpful to show the baseline characteristics of the participants.

Thank you for this comment. We modified the first paragraph of the Methods section accordingly, by providing the number of human participants and mice and we have included a new table with the patients' characteristics (Supplementary table 1).

Round 2

Reviewer 1 Report

I thank the authors for the response to the comments. I have no further comments.